# Thermal vulnerability of sea turtle foraging grounds around the globe
Forough Goudarzi [1] ✉, Aggeliki Doxa[2,3,4], Mahmoud-Reza Hemami [5] & Antonios D. Mazaris[2]

Anticipating and mitigating the impacts of climate change on biodiversity requires a comprehensive understanding on key habitats utilized by species. Yet, such information for high mobile marine megafauna species remains limited. Here, we compile a global database comprising published satellite tracking data (n = 1035 individuals) to spatially delineate foraging grounds for seven sea turtle species and assess their thermal stability. We identified 133 foraging areas distributed around the globe, of which only 2% of the total surface is enclosed within an existing protected area. One-third of the total coverage of foraging hotspots is situated in high seas, where conservation focus is often neglected. Our analyses revealed that more than two-thirds of these vital marine habitats will experience new sea surface temperature (SST) conditions by 2100, exposing sea turtles to potential thermal risks. Our findings underline the importance of global ocean conservation efforts, which can meet climate challenges even in remote environments.

A growing amount of evidence is unveiling an emerging marine biodiversity crisis[1,2]. The systematic assessments of the International Union for Conservation of Nature (IUCN) classify more than 1500 marine species, as Critically Endangered, Endangered, or Vulnerable[3,4], while more than 110 local[5] and 20 global extinctions have been reported by IUCN[3]. These numbers are even more alarming if we consider that only a portion (less than 8%) of marine taxa have been assessed by now[4], while models projecting climate change impacts in marine biodiversity suggest that by the year 2100, more than 90% of the assessed species will face an increasing risk of extinction, with the majority of species subjected to shrinkages of their suitable habitats[6]. Under this context of accelerating marine biodiversity loss and the uncertainty that future climate causes, it is now essential to take immediate action. Yet, the first step in order to conserve marine biodiversity is to delineate the spatial distribution of at-risk species[7] and estimate their vulnerability to pronounced current and future threats[8].

For species with restricted geographic distributions that utilize a number of distinct, spatially limited habitats, the assessment of their vulnerability could enclose information both on different life stages and on the state of the exact habitat used by the species throughout its life history[9,10]. This is, however, not the case for wide-ranging marine species, which undertake long migrations and utilize various habitats even on an annual basis[11]. Sea turtles are a classic example of a highly migratory group for which assessments of their status are based on fragmented information

mainly derived from the nesting sites hosted on sandy shores, where females emerge once a year to lay their nests[12]. While there have been a number of studies where sea turtles have been satellite tracked (e.g., reviewed in ref. 13), there have been very limited efforts to collate all these accumulated data to examine the overall threats to sea turtles globally.

For sea turtles, migration to breeding sites does not occur in successive years, and depends on the provision of sufficient fat as fuel gained in the foraging grounds[14,15]. Therefore, the conditions at the foraging habitats determine the number of turtles which could reproduce[16], and thus could be responsible for variations in population trends and differences in the status of different populations[17]. As the wealth of foraging grounds largely depends on thermal conditions, it is likely that an altered climate could degrade the quality and availability of dietary sources[18], decreasing their carrying capacity[19], and could further result in weak body conditions[16], subsequently negatively affecting overall population productivity. Indeed, the lack of a comprehensive and integrated perception of foraging habitats has postponed an international action plan on the conservation of multinational distribution ranges of sea turtles[20,21].

Here, we compile a global database of satellite tracking records to spatially delineate key foraging grounds for the seven extant species of sea turtles. We also establish an analytical framework to examine the thermal stability of these foraging grounds against sea surface temperature (SST) projections. Our analysis delineated 133 key foraging areas for the seven

[1]Department of Biodiversity and Ecosystem Management, Environmental Sciences Research Institute, Shahid Beheshti University (SBU), Tehran, Iran. [2]Department of Ecology, School of Biology, Aristotle University of Thessaloniki, Thessaloniki 54124, Greece. [3]Institute of Applied and Computational Mathematics, Foundation for Research and Technology-Hellas (FORTH), Heraklion, Crete, Greece. [4]Department of Biology, University of Crete, University Campus Vouton, 70013 Heraklion, Greece. [5]Department of Natural Resources, Isfahan University of Technology, Isfahan 8415683111, Iran. ✉e-mail: f_goudarzi@sbu.ac.ir

species worldwide. Our findings indicate that the majority of these foraging areas do not overlap with marine protected areas (MPAs). Currently, less than 2% of the identified hotspots are fully covered by existing MPAs, while one-third of the total foraging hotspots coverage is located in the high seas, where meeting conservation targets might be even more challenging. Additionally, our analysis indicates that more than two-thirds of these critical habitats will be exposed to novel temperatures by the end of the century. Our findings regarding the thermal vulnerability of these predominantly unprotected and often distant habitats carry significant implications for global ocean conservation practice.

## Results

We spatially delineated sea turtles' foraging grounds, using 4817 locations of foraging individuals (Supplementary Fig. 1), derived from satellite tracking data ($n = 1035$) published in scientific literature. To summarize the spatial distribution of these locations, non-parametric kernel density estimates were derived using region-specific bandwidths to account for variations in the reported point data. We identified 133 foraging hotspots which are distributed between latitudes 50° N and 40° S (Fig. 1 and Supplementary Table 1).

A significant relationship was identified between the coverage of foraging hotspots and latitude ($p < 0.05$), primarily influenced by the notably larger foraging grounds of leatherback turtles observed in the southern regions (Supplementary Fig. 2). To investigate the proximity of foraging grounds to the coastline, we calculated the Euclidian distance from the shore using the centroids of each foraging polygon. The vast majority of foraging grounds (i.e., 79 out of 133) were located within a 100 km distance from the shoreline (Fig. 1). However, the distances varied significantly among the studied species (H = 55.133, $p < 0.01$). On average, the foraging grounds of green turtles were found to be significantly closer to the coast, as defined by Dunn's method, compared to leatherbacks and olive ridley sea turtles (in both pairwise comparisons, $p < 0.01$). The most distant foraging grounds were identified for leatherbacks and loggerheads in the mid-Atlantic Ocean and northwest Pacific Ocean, respectively. Our analyses revealed a significant association between the size of a foraging area and its distance from the coast ($p < 0.01$). Limited in number ($n = 14$ out of 133) yet large in surface foraging grounds were identified in high seas, representing 32.6% of the total coverage of foraging grounds, including 47.6 and 36.7% of the foraging surfaces of leatherbacks and loggerheads, respectively.

To assess the inclusion of these foraging areas within existing Protected Areas, we compared their boundaries with the global MPA network obtained from the World Database of Protected Areas[22]. Our analysis revealed that only 2% of the total surface of sea turtles' foraging hotspots fall within existing MPAs, while 57% of their total surface is completely unprotected (with less than 5% of their extent covered by an existing MPA)

(Fig. 2 and Supplementary Fig. 3). Among the seven sea turtle species, the foraging grounds of flatbacks exhibited a relatively higher level of protection coverage, with 36% of the identified sites' surface being subjected to protection coverage exceeding 50% (Fig. 2). However, this was not the case for all six remaining species, as only a slight portion (0–7%) of their foraging distributions adequately overlapped with the current MPA network (Fig. 2 and Supplementary Fig. 3).

To explore the potential thermal stability of each of the foraging grounds, we used a time series of monthly minimum and maximum SSTs and calculated the dissimilarity of temperature distributions for present and future periods by means of the Hellinger distance. The analysis revealed that by the end of the century, more than two-thirds (68.6%) of the total foraging habitat coverage across species, will be exposed to novel temperatures (Fig. 3). Thermal novelty (TNo) values ranged from 0, for the absence of novel conditions in the future compared to the present ones, to 1 for complete novelty, with TNo ≥0.5 values indicating a moderate degree of novel thermal conditions in the future, while TNo ≥0.8 indicated areas with almost alien thermal conditions[23]. High TNo values were overall projected for sea turtles' foraging habitats that are located in latitudes around the equator (Fig. 3a), with relatively lower TNo when moving to poleward habitats, while no trend was observed over the longitude gradient (Supplementary Fig. 4).

The highest risk of exposure to thermal novelty was projected for the foraging areas of hawksbill and olive ridley sea turtles, with median thermal novelty values of 0.66 [0.31–0.98] and 0.63 [0.38–0.92], respectively (Fig. 3b), with 83.1 and 98.2% of the species respective habitats expected to have a moderate degree of thermal novelty (TNo ≥0.5). The foraging grounds of leatherback and green sea turtles were also expected to be highly exposed to novel temperature conditions, with median TNo values of 0.53 [0.17–0.90] and 0.54 [0.33–0.86], respectively (Fig. 3b), and with 69.2 and 68.9% of the species habitats respectively, experiencing thermal novelty by 2100. Almost alien thermal conditions (TNo >0.8) are particularly projected for the olive ridley sea turtles covering 42.2% of the species foraging habitats, located at the northeast coasts of South America. Alien thermal conditions were also projected for hawksbill, leatherback, and green sea turtles but to a lesser extent, covering 4.8, 11.1, and 16% of the species respective foraging habitats, situated in the Pacific Ocean and the northeast coasts of South America for hawksbill sea turtles and in the Pacific and Indian Ocean for the green and leatherback sea turtles (Fig. 3c).

The lowest exposures to novel temperature conditions were projected for Kemp's ridley and loggerhead sea turtles, with median TNo values of 0.27 and 0.45 [0.29–0.87] respectively (Fig. 3b) and with 0% and 28.2% of their respective habitats having TNo values ≥0.5, while medium deviation from current temperatures is expected for the flatback sea turtles' habitats, for which a median TNo value of 0.52 [0.42–0.70] was projected, with 71.2%

**Fig. 1 | The worldwide distribution of key foraging hotspots of adult sea turtles.** Polygons indicate the boundaries of 50% isopleth of the kernel density estimation of foraging grounds. Hotspots are distributed between latitudes 50° N and 40° S, mainly located within a 100 km distance from the shoreline.

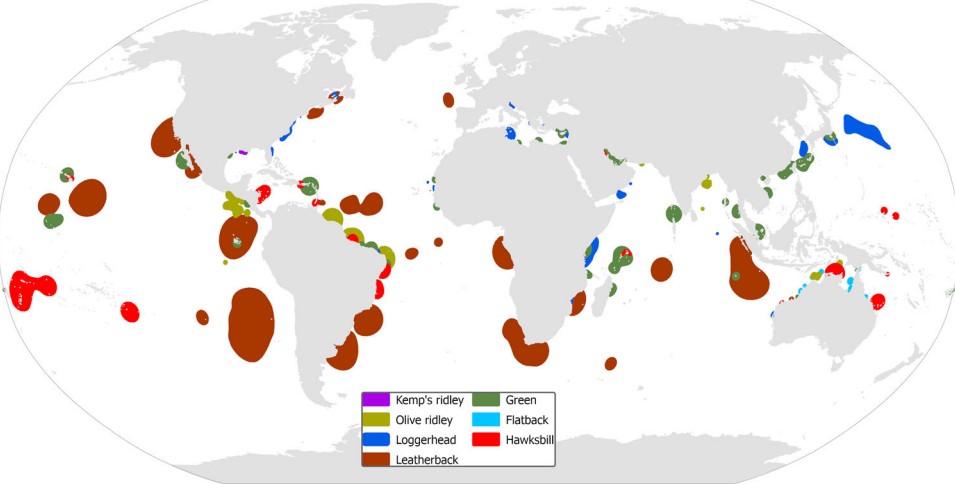

**Fig. 2 | The percentage coverage of sea turtles' foraging grounds by the current network of marine protected areas (MPAs) per species (bars) and for all turtle species combined (pie chart).** Less than 2% of global foraging grounds are covered by MPAs.

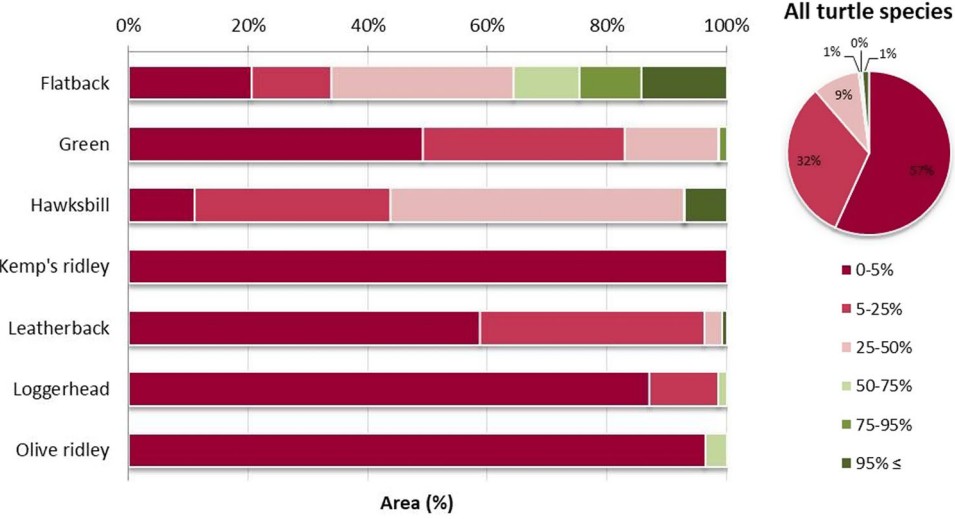

**Fig. 3 | Thermal novelty (TNo) within sea turtles' foraging hotspots by 2100. a** TNo over latitude, **b** TNo per sea turtle species, and **c** the spatial pattern of TNo within foraging hotspots. Polygons indicate the boundaries of 50% isopleth of the kernel density estimation of foraging grounds.

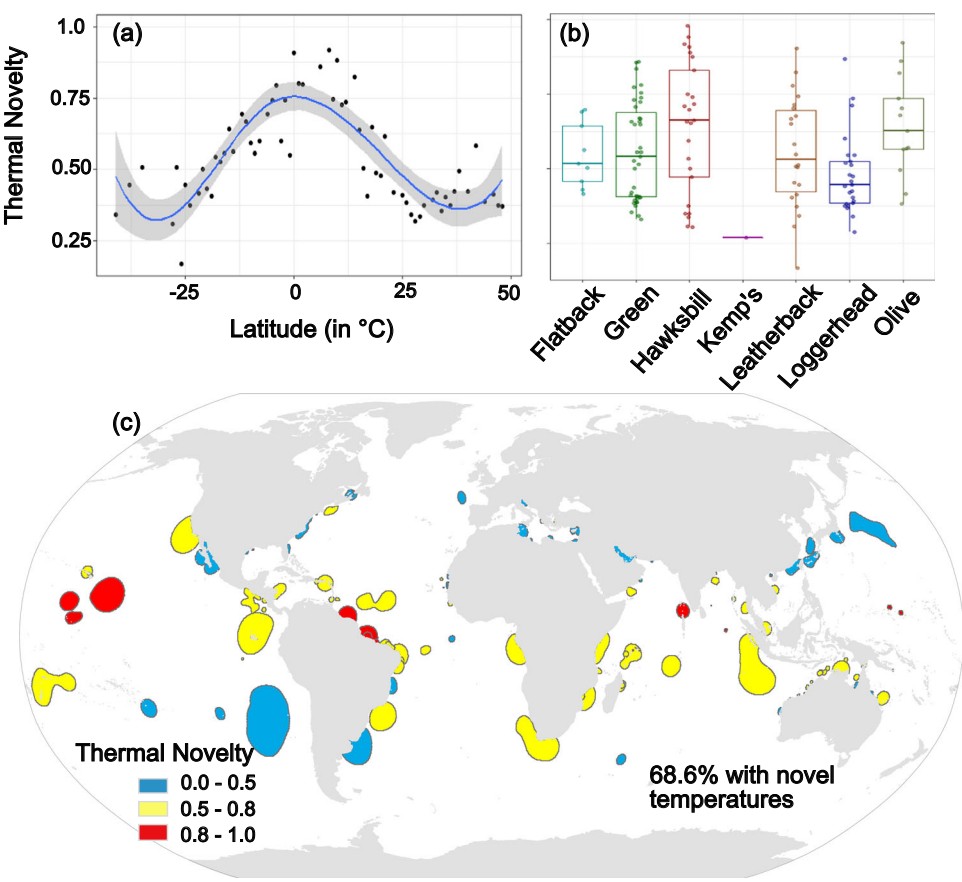

of the species foraging hotspots expected to experience some thermal novelty (TNo >0.5) but no completely alien thermal conditions (TNo <0.8) (see Supplementary Fig. 5 for TNo maps per species).

## Discussion

Identifying key marine habitats for highly mobile species and evaluating their risk from climate change could support the development of comprehensive management and conservation strategies[24,25]. Here, we present the first, global dataset of foraging sites for the seven sea turtle species, highlighting challenges related to the current protection coverage of these key habitats and their remote locations. The outputs of this study represent an opportunity for planners aiming to expand MPAs to 30% of the oceans in order to meet the global target of the Convention on Biological Diversity by 2030[26].

Oceans will experience novel or even completely alien temperature conditions in the future[23]. Indeed, global warming accelerates the range shifts of marine species approximately ten times faster than terrestrial organisms[27]. We revealed that more than two-thirds of the global coverage of sea turtles' foraging habitats will experience novel sea surface temperatures by the end of the century. The sea turtle species that forage in tropical areas, characterized by higher SSTs, are generally expected of being exposed to novel temperature conditions in the future. Still, many foraging grounds

in subtropical areas are likely to be subjected to significant deviations in their current temperatures. In some cases, such as the foraging grounds of loggerhead turtles in northern sites of the Mediterranean, even moderate SST changes might be hard to counterbalance, as the land masses hinder a northward expansion of current distributions[28].

Sea turtles' body temperature is intrinsically influenced by ambient temperatures, making them susceptible to rapid climate change[29]. Although their endothermic ability permits them to regulate their body temperature in cold water, often enabling them to forage in cool, prey-rich northern grounds, they have less thermal regulatory capacity in warmer water[30]. To cope with increasing temperatures, species are forced to spatially shift their distributions to climatically more suitable habitats[18,28]. In that context, hawksbills temporally migrated from existing foraging grounds to cooler and deeper regions to avoid extreme summer temperatures[31]. Similarly, during an El Nino event in the eastern Pacific Ocean, olive ridley sea turtles responded to changing ocean temperatures by moving northward to forage in upwelling grounds[32]. It has also been documented that loggerhead breeders shifted to cooler foraging grounds, as warmer oceans support lower productivity and prey abundance, which can have time-lagged effects on sea turtles breeding capacity during the following years[33]. While physiological and behavioral mechanisms, such as diving in deeper waters, migration, nocturnal foraging, and dormancy or hibernation-like status may permit species to avoid adverse conditions, other characteristics, such as foraging site fidelity[34] can amplify their exposure to climate change risks.

Climate change is actively restructuring and redistributing marine communities across trophic levels[35] from producers to mega-consumers such as sea turtles. Species sensitivity to climate change can also be influenced by an animal's diet strategy[36], while the complex trophic relationships of the marine realm could affect prey availability or shortages in the future[37]. In terms of dietary strategy, omnivorous species, like flatback, loggerhead, Kemp's ridley, and olive ridley sea turtles, are probably less prone to prey shortages than species with narrow or specific diets, such as green turtles[16]. Green turtles are highly dependent on seagrass pastures and are probably more sensitive to warming than carnivore species[38]. Previous studies[39] have predicted a significant habitat loss of seagrass under global warming pressure, especially in the central Indo-Pacific Ocean, where the green turtles' foraging hotspots will experience a completely alien climatic condition. Hawksbills also have narrow dietary preferences, being mainly spongivorous, which makes them particularly sensitive to direct and indirect climate change impacts[40]. Although the responses of sponges to climate warming and related ocean acidification are still unclear, with possibly even positive effects, especially for phototrophic and bioeroding species[41], the complex interactions between prey availability and sea turtles' thermal tolerance will still be a limiting factor in habitat use and potential shifts.

Our study highlighted a significant gap in the protection of sea turtles foraging hotspots at the global scale, as less than 2% of the foraging grounds were completely covered by the current MPA network. As the majority of established MPAs are adjacent to the coastline, with only a few sites located outside the neritic zone in open sea habitats[42], it seems that foraging populations that depend on near coastal habitats might be better protected, while others that forage in distant habitats from the coastline can be considerably neglected. Indeed, our analysis showed that the main foraging areas of flatback and hawksbill turtles are located within the neritic zone, possibly justifying why they are relatively better included in existing MPA networks, while the percentages of foraging habitats under protection for the remaining species remain very low or null. However, considering that multiple foraging areas are situated at greater distances from the coast, and tend to expand as we move farther away, we underscore the importance of exploring ways to enhance our conservation efforts in the open ocean. Still, recognizing that many countries have implemented laws to safeguard sea turtles, it's important to acknowledge that populations still face increased mortality risks, such as bycatch or collisions with boats. Despite being protected in these countries, whether within a designated MPA or outside one, sea turtles remain vulnerable. While we appreciate the significance of MPAs in bolstering sea turtle protection, safeguarding this highly mobile

species demands a more comprehensive conservation approach. Within this context, delineating key habitats spatially and designing important marine turtle areas becomes imperative.

The present analysis, by reviewing and mapping available information on tracked sea turtles and by providing a complete and exhaustive mapping of their foraging habitats at the global scale, offers a background for effectively considering critical sea turtles' hotspots, even in deep-sea habitats, in our conservation planning. Our findings could therefore serve as a first step to schedule conservation actions in light of the potential risk of global warming for foraging hotspots. Nevertheless, we emphasize the necessity of climate-smart systematic conservation planning for sea turtles at the global scale. This approach could allow for the incorporation of species complex biology, considering both terrestrial and marine habitats that sea turtles depend on, while accounting for the three-dimensional nature of the ocean environment[43].

Still, we caution that other significant foraging areas (e.g., ref. 44) exist; nevertheless, the information regarding their distribution does not conform to the methodological criteria applied in this study. Also acknowledging that sea turtles are not tracked from every nesting site, and that individuals from the same nesting site might visit different foraging areas, it is important to expand our approach to include various sources of information. For example, aerial surveys and stranding data can help supplement tracking data to highlight key areas[45]. We further want to emphasize that although our method provides a spatial delineation of key foraging areas, the precision of the derived locations relies heavily on the quality of the gathered data. Specifically, the accuracy of the digitalized foraging points used to identify these areas might fluctuate based on the accuracy of the source maps we employed. Hence, it is imperative for future research to prioritize the use of original tracking data, aiming to mitigate potential spatial biases as much as possible. Finally, the use of other than temperatures, climatic variables, and multiple climatic models could further enhance the outcomes of this study, providing a possible spectrum of climate risk scenarios and enabling further uncertainty analyses.

Our findings underline the need for coordinated, international efforts in upcoming marine conservation planning to align with the objectives of the Convention on Biological Diversity for 2030. We showed that the majority of sea turtles foraging hotspots, as important habitats that could host a high number of reproductive individuals, are spatially unprotected and largely exposed to ocean warming. Moreover, we highlight that one-third of these habitats are located on high seas, outside countries' exclusive economic zones, where meeting conservation targets and climate resilience becomes even more challenging[46,47]. We conclude that failing to design and implement conservation plans that can effectively and strategically include climate-resilient areas, even in remote, high-seas habitats, may have detrimental effects on sea turtles at a global scale.

## Methods
### Data on foraging sites
To ensure that we could build our spatial analysis upon standardized information on foraging habitat use, we explored locations of foraging animals obtained through satellite telemetry data. Satellite telemetry data offer precise spatial information obtained based on the same principles and technological characteristics, and even though they might be subjected to potential biases (e.g., tagging location, sample size, data gaps, and processing), they are considered key sources of information for delineating habitat use of highly migratory species[48]. We performed a literature search on Google Scholar using the terms sea turtle or marine turtle, satellite telemetry, and foraging. Our research included scientific articles, and gray literature (i.e., reports, theses, and proceedings of conferences) that were published from 1982 to 2020. Once these publications were scanned and reviewed, we only selected those ($n = 213$) that tracked adult turtles between their foraging habitats and breeding sites.

For our analysis, we maintained only complete tracks ($n = 1035$), excluding the cases for which the transmitter might have failed before arrival at foraging locations. To identify potential foraging sites for pelagic foragers,

we examined shifts in their horizontal behavior. Our focus was on locations where transmitted signals clustered within close proximity and instances where we observed substantial changes in the direction of subsequent transmissions[49]. We georeferenced and digitized all tracks collected. The georeferencing process encompassed four main steps. Initially, we gathered the relevant map/s from each article. Subsequently, we pinpointed control points (i.e., distinct and easily recognizable geographic elements) on both the extracted map and a georeferenced map available within the GIS framework. For the third step, these control points served as reference markers to align the extracted map precisely with the georeferenced map in GIS. This alignment process involved meticulous adjustments in position, rotation, and scale to ensure accurate alignment between the two maps. Finally, we attributed coordinates to the extracted map by leveraging the known coordinates from the georeferenced map, utilizing the control points as anchors that connect and synchronize the two maps. After digitalizing the tracks, we extracted a minimum of one point to signify a foraging location from each animal track. These points were assigned either at the beginning or end of each trajectory. In instances of complex tracks involving multiple stops or significant directional changes, we selected multiple points. Generally, a foraging point was marked whenever a notable change in trajectory direction was observed in successive transmissions. This method of selecting multiple points aimed to more accurately delineate the extent of foraging grounds in those tracks. Overall, sea turtles' adult foraging locations were represented by 4817 points in the dataset, of which 262 corresponded to hawksbill turtles, 877 to green turtles, 1407 to loggerhead turtles, 128 to flatback turtles, 1135 to leatherback turtles, 466 to olive ridley sea turtles, and 542 to Kemp's ridley turtles (Supplementary Fig. 1). These locations were identified within 54 of the 58 regional management units of sea turtles, which represent discrete broad geographic areas based on genetics, distribution, movement, and demography[50].

### Identifying foraging hotspots
We extracted and georeferenced the locations of foraging adult turtles from the identified sources. To illustrate the post- or pre-nesting migration on maps, we defined the end or start point of each tracked individual's route to the foraging area as the designated foraging site. Further details on the number of tracks and the extracted points are given in the Supplementary Data.

We considered the centroid of the 50% data distribution isopleth, generated through kernel density estimations, as representative of the high-use area (see also refs. 51, 52). To estimate the kernel density, we used the Kernel density tool from the Spatial Analyst extension in ArcGIS v.10.6.1. First, the point data related to the foraging sites of each species were clustered based on sea turtle regional management units (RMUs[50]); so that the foraging hotspot could be identified for each RMU separately. The kernel bandwidth was chosen regionally to account for variations in the density of the point data. Such a variable-bandwidth approach smooths the tails and gets high resolution in regions with high-density points[53]. The cylindrical equal-area projection was used for all geospatial analyses.

### Distance to coastline
Aiming to explore the proximity of the foraging grounds to the shore, we calculated the distance of the centroid of each polygon to the coastline. We used the High-resolution Shoreline (GSHHS) dataset (NOAA; available at: www.ngdc.noaa.gov/mgg/shorelines/gshhs.html), which provides full resolution boundary between land and ocean.

### Protection coverage and high seas
We investigated whether the current marine protected area network adequately covers the sea turtles' foraging grounds, by overlaying their distribution with the spatial extent of the marine protected areas. The boundaries of the protected areas were derived from the World Database of Protected Areas[22].

We also determined the proportion of foraging hotspots whose centroids are located on the high seas, by overlaying their distribution with a map of the exclusive economic zones (EEZs), derived from www.marineregions.org.

### Thermal novelty index (TNo)
Daily data regarding the thermal range of the foraging habitats were retrieved based on the CMIP6 at the finest possible resolution, at 0.25° using the GFDL-CM4 model. In order to couple the identification of foraging areas analysis with the thermal novelty analysis, we specifically required the finest resolution in daily temperature data, and no other climatic model with comparable detail was available. 2000–2014 was set as the present baseline period, while for future projections, the end of the century (2085–2100) was considered, based on the spp585 scenario. Average min, max, and mean SSTs of the foraging areas per species for the baseline and the future time periods are shown in Supplementary Table 2. Time series of monthly minimum and maximum temperatures were then calculated over the baseline and future periods. To compute the dissimilarity of baseline (P) and future (Q) temperature variable (j) for k months, the Hellinger distance (HD) was used: where HD was calculated for each cell (x) included in each foraging habitat. Then the average HD values of the two climatic variables (min and max temperatures) were computed per cell, representing the overall thermal novelty (TNo) of that cell in the future. In order to determine the TNo index per foraging habitat, we averaged all grid cells TNo values that were located within each habitat's boundary. The Hellinger Distance is a statistical metric used to quantify the similarity or dissimilarity between two distributions[54], in this case, present and future SST distributions. It is bounded between 0 and 1, with 0 indicating identical distributions, thus no thermal deviation between present and future temperatures, and 1 indicating completely disjoint distributions, thus indicating completely novel thermal conditions. A Hellinger Distance value of 0.5 (or higher) generally indicates a moderate dissimilarity between the two distributions. The percentage of areas experiencing novel conditions out of the total foraging areas where also estimated.

### Statistics and reproducibility
The minimum distance between the centroid of each of the 133 foraging grounds and the coastline was estimated using the Near tool in ArcGIS v.10.6.1. We performed a Kruskal–Wallis analysis to assess potential differences in the distance of foraging grounds from the coastline among various species. Additionally, we conducted pairwise comparisons of these distances between each species using Dunn's method. We should note that the former analysis was not performed for Kemp's ridley, for which only one foraging area was identified. To assess whether larger foraging grounds are situated in more distant areas, we applied a Spearman correlation coefficient; this involved comparing the log-transformed size of the foraging areas with their respective distance from the shore. We also used linear regression to address the relationship between the size of foraging areas and latitude. The smooth trend line in Fig. 2a was fitted using the ggplot2[55] package with the linear model (lm) method and a smoothing span of 5, while the gray zone represents the 95% confidence interval. All statistical analyses were conducted in R v.4.3.2, and statistical significance was indicated for $p$ values <0.05. The climatic analyses were also performed in the R environment following the method developed by ref. 23. All maps were produced in ArcGIS v10.6.1.

### Reporting summary
Further information on research design is available in the Nature Portfolio Reporting Summary linked to this article.

### Data availability
The raw data files required to reproduce the analyses are available as Supplementary Data files directly uploaded with the journal. All other data are available from the corresponding author on reasonable request.

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

## Acknowledgements
This work is based upon research funded by Iran National Science Foundation (INSF), under grant number 4000758.

## Author contributions
A.D.M. conceived the original idea and coordinated the study. F.G., A.D.M., and M.R.H. designed the study. M.R.H. provided resources. F.G. collected and synthesized data. F.G., A.D., and A.D.M. conducted the analyses. F.G. and A.D. wrote the initial manuscript. All authors contributed to the interpretation of the results, writing and reviewing of the manuscript.

## Competing interests
The authors declare no competing interests.
