## [Peer Review File · Communications Biology]

Reviewers' comments:

Reviewer #1 (Remarks to the Author):

Here the published tracks of almost 5000 turtles are digitised and then the proportion of foraging sites within marine protected areas is examined. The authors conclude that few turtle foraging sites are within protected areas and discuss the concerns surrounding this finding.

This is a nice manuscript that I enjoyed reading. The authors do an impressive job collating a lot of the sea turtle tracking literature. This is a huge task and allows a global analysis of the overlap of foraging grounds with marine protected areas. This global analysis is very novel and will appeal to a broad audience. The analysis and conclusions are very clear. There are some caveats with the analysis (see below) but these can all be discussed and do not detract from the key conclusions. I think this will make a nice contribution and I only have few relatively minor suggestions.

1.

Line 46. "Still, information on the spatial extent or status of the foraging grounds where animals spend the vast majority of the life is often scarce. Indeed, we even have a very limited knowledge on the global location of these key habitats"

Line 57. "Still, regardless of the key role of foraging grounds on sea turtle persistence, our actual knowledge on the location of these critical habitats remains very limited"

I think these statements are incorrect. There has been a huge amount of work satellite tracking sea turtles. See for example, Figure 1 in "Satellite Tracking Sea Turtles: Opportunities and Challenges to Address Key Questions. doi: 10.3389/fmars.2018.00432". I think you need to say that while there have been a huge number of studies where sea turtles have been satellite tracked (e.g. reviewed in the above), there have been very limited efforts to collate all these accumulated data to examine overall threats to sea turtles".

2. Line 82. "No association was revealed between the surface of foraging hotspots and latitude ($p > 0.05$)."

I do not understand. Is a word missing after "surface" ?

3. I think that somewhere you need a few lines to acknowledge that you have likely missed some key foraging areas. For example, in Figure 1 it looks like you have massively underestimate the high seas foraging areas for leatherbacks. See for example, Figure 3 in "Fossette S et al. (2010). Spatio-temporal foraging patterns of a giant zooplanktivore, the leatherback turtle. Journal of Marine Systems 81, 225–234. doi:10.1016/j.jmarsys.2009.12.002". You completely miss all the North Atlantic oceanic foraging sites.

Likewise you miss all the classic work with tracking leatherbacks in the eastern Pacific. See: "Shillinger GL, Palacios DM, Bailey H, Bograd SJ, Swithenbank AM, et al. (2008) Persistent leatherback turtle

migrations present opportunities for conservation. PLoS Biol 6(7): e171.
doi:10.1371/journal.pbio.0060171”

Likewise you miss all the classic work tracking leatherbacks from the West Pacific. See: “Benson, S. R., T. Eguchi, D. G. Foley, K. A. Forney, H. Bailey, C. Hitipeuw, B. P. Samber, R. F. Tapilatu, V. Rei, P. Ramohia, J. Pita, and P. H. Dutton. 2011. Large-scale movements and high-use areas of western Pacific leatherback turtles, *Dermochelys coriacea*. *Ecosphere* 2(7):art84. doi:10.1890/ES11-00053.1”

Likewise the classic work tracking loggerheads from their largest rookery in the world, see:
“Hawkes et al. Phenotypically Linked Dichotomy in Sea Turtle Foraging Requires Multiple Conservation Approaches. *Current Biology* 16, 990–995. DOI 10.1016/j.cub.2006.03.063”

I appreciate you cannot capture everything, but these are really high-profile studies that have been missed. I could make a list of 50 really important papers that have been missed. I suggest you add some words along the lines of “Our study is a first attempt to compile data from across many different studies and we appreciate that important studies may have been missed through our key word searches. For example, important studies showing pelagic foraging hotspots by leatherback and loggerhead turtles (e.g. Fossette et al, Shillinger et al, Benson et al., Hawkes et al.) were not included”.

4. I suggest you likewise add a couple of lines to say that moving forward it is also important to incorporate foraging ground distribution data from a variety of sources. For example, turtles will not have been tracked from every nesting site. Other sources of important information include surveys on foraging grounds (e.g. aerial) and stranding data and often these can help supplement tracking data to highlight key areas. e.g. see many examples including “A review of the importance of south-east Australian waters as a global hotspot for leatherback turtle foraging and entanglement threat in fisheries. *Marine Biology* 170: 74. <https://doi.org/10.1007/s00227-023-04222-3>”

5. Methods. Usually the method to delineate a foraging site is that turtles are tracked to a location where they then remain for many months. I would add this detail to your methods, so it is clear to reader that you did not include incomplete tracks where the transmitter failed before the turtle arrived at its destination.

6. Methods For pelagic foragers how did you assess when the foraging grounds were reached versus animals still migrating ?

7. Line 237. “We georeferenced and digitized all areas that were identified as adult foraging sites.” In a line or two I would explain how you did this georeferencing and the likely resolution ... e.g. accurate to a few 10s of km or better ?

7. I think you need to discuss that there are many routes to sea turtle conservation. For example, in many countries it will be illegal to hunt, harm or kill sea turtles, regardless of where they are found. So in many countries turtles are protected regardless of whether they are in an MPA. I think you need to make

this statement but then explain that while national laws are often in place to protect sea turtles, they may still suffer mortality, e.g. through bycatch or boat strikes. So a holistic view of needed of conservation. It is not just about MPAs. I think this is a key issue that is often ignored.

8. In answer to the specific journal questions:

Does the manuscript have technical or conceptual flaws that should prohibit its publication?

No

Are the conclusions original?

yes

Do you feel that the results presented are of immediate relevance for people in your own discipline or for a broader audience?

For a broader audience – see above,

In summary, a nice manuscript that I look forward to seeing published.

Reviewer #2 (Remarks to the Author):

Overall, I think this is a nice assessment of the potential change in thermal conditions for sea turtle foraging sites. Its great to have this comprehensive analysis in one study. I think, however, its a bit of a stretch to refer to this as a 'Climatic vulnerability' when only having discussed thermal conditions. Also need to specify which thermal variables were used in this study. The study is novel in that it encompasses all turtle species and provides a global assessment. It would be more impactful to include additional variables beyond 'thermal conditions' to make this a true study of climate conditions, or include more climate models beyond CM4 to show the variability in the CNo values for each the foraging sites.

Devising the CNo value is also a nice novel method of making it easier to compare foraging site changes between regions and species. As a result, I think this manuscript will have value to the field.

Line 27: Change first sentence to 'A growing **amount** of evidence...'

Lines 47 – 49: This sentence is hard to interpret. Please rewrite to clarify.

Lines 50- 51: This sentence needs a citation.

Line 59 – 61: This sentence needs a citation.

Line 63 – 65: This is an oversell of your study. You specifically only worked with temp data (and which temp data needs to be clarified), so to say you examined 'climate' stability is inaccurate. You examined temperature stability, and you need to specify which temp (SST, air temp, temp at a depth, ect...).

Line 69 – 71: Again, 'novel climates' is inaccurate, you mean 'novel temperatures'. This needs to be changed throughout the manuscript.

Line 82: It's unclear what is meant by the 'surface' of a foraging hotspot.

Lines 106 – 108: Again can't use climatic so loosely, and need to define which temp variables used.

Lines 110 – 114: This explanation of the CNo values is repeated from the methods.

Lines 117 – 138: This is good details, but I think you can also include a little more information on the actual thermal habitats. For each species include the current thermal conditions of their foraging grounds (temp and variability) and then the future conditions to help demonstrate what is meant by high and low novelty of the future conditions. It's very hard to interpret these CNo results without any actual information about the environments you are referring to.

Line 118: Correct to 'olive **ridley** sea turtles'. Use the complete name throughout, not simply 'olive sea turtles'.

Lines 145-147: This sentence needs a citation. Correct spelling of 'Convention'.

Line 151: You have not actually studied 'suitability', so this sentence is inappropriate based on your results. Either delete or clarify.

Line 151-152: This sentence is worded strangely. Is it meant to say "The tropical zone is the most likely region to have novel climate conditions in the future"? Please clarify.

Line 154 – 156: This is assuming northward expansion is the only option for loggerheads. They can also leave the Med. Add a citation to specify your point or clarify this sentence.

Lines 157 – 158: This sentence needs a citation.

Lines 160– 162: This sentence needs a citation.

Lines 163 – 164: Number this citation.

Line 188: Sentence ending in this line needs citation.

Lines 215 – 216: I like this concept of helping to 'schedule conservation actions'. Perhaps include a figure/table of your recommendations for which foraging site for each species should be protected first.

Lines 236 – 237: You need to provide the citations for all of the publications you used for defining/mapping foraging grounds. Perhaps a supplemental table.

Lines 284 – 301: Overall the CNo section needs more detail. Why did you choose CM4 vs other climate models? Perhaps worth using a few models with similar resolutions to see if there is variability in calculating the CNo index for each foraging site. You also only incorporate temp, so calling it a 'Climatic novelty index' is overselling it. It's a 'Thermal novelty index.' You need to specify which thermal variable was used – SST, air temp, temp at some depth? You cannot simply say 'thermal conditions' without specifying which temp variable was used, and then you need to clarify this throughout the manuscript. Need a better definition of the Hellinger Distance. For example, "with CNo \geq 0.5 values indicating some degree of novel thermal conditions in the future" This is far too vague. What is meant by 'some degree'. It is hard to interpret the change you are trying to demonstrate without some tangible value for this. Is 'some degree' equal to a specific temperature range shift? What change in temp is equal to 'complete novelty'?

Fig 1, Fig 3, Fig S1, and Fig S2: For all maps, perhaps crop out Antarctica and portions of the Arctic and zoom in a bit.

I think an interesting additional set of maps would be to show the CNo values for each foraging ground by species. This can be supplemental info, but I think it would be helpful for researchers who are more focused on certain species over other. Having these maps would be very helpful in providing more in depth information for Fig3a and b.

Table S1: Perhaps include some summary of the 'thermal conditions' (again is this SST, air temp, temp at some depth) as separate columns – min, max, average, time of year.

Response to Reviewer

Reviewer #1

1st Comment

Line 46. “Still, information on the spatial extent or status of the foraging grounds where animals spend the vast majority of the life is often scarce. Indeed, we even have a very limited knowledge on the global location of these key habitats”

Line 57. “Still, regardless of the key role of foraging grounds on sea turtle persistence, our actual knowledge on the location of these critical habitats remains very limited”

I think these statements are incorrect. There has been a huge amount of work satellite tracking sea turtles. See for example, Figure 1 in “Satellite Tracking Sea Turtles: Opportunities and Challenges to Address Key Questions. doi: 10.3389/fmars.2018.00432”. I think you need to say that while there have been a huge number of studies where sea turtles have been satellite tracked (e.g. reviewed in the above), there have been very limited efforts to collate all these accumulated data to examine overall threats to sea turtles”.

Our response: Thank you for pointing this out. We revised the text according to your suggestion. The revised text (lines 45-51) now reads:

‘Sea turtles are a classic example of a highly migratory group for which assessments on their status are based on fragmented information mainly derived from the nesting sites hosted on sandy shores, where females emerge once a year to lay their nests¹². While there have been a number of studies where sea turtles have been satellite tracked (e.g., reviewed in ¹³), there have been very limited efforts to collate all these accumulated data to examine the overall threats to sea turtles globally.’

2nd Comment

Line 82. “No association was revealed between the surface of foraging hotspots and latitude ($p > 0.05$).”

I do not understand. Is a word missing after “surface” ?

Our response: Thank you for noting this out. In the revised ms, we included more foraging sites into the analysis and the relationship between area and latitude of foraging grounds became significant. The revised text now reads (see lines 87-89):

‘A significant correlation was identified between the surface area of foraging hotspots and latitude ($p < 0.05$), primarily influenced by the notably larger foraging grounds of leatherback turtles observed in the southern regions (Figure S2).’

3rd Comment

I think that somewhere you need a few lines to acknowledge that you have likely missed some key foraging areas. For example, in Figure 1 it looks like you have massively underestimate the high seas foraging areas for leatherbacks. See for example, Figure 3 in “Fossette S et al. (2010). Spatio-temporal foraging patterns of a giant zooplanktivore, the leatherback turtle. Journal of Marine Systems 81, 225–234. doi:10.1016/j.jmarsys.2009.12.002”. You completely miss all the North Atlantic oceanic foraging sites.

Likewise you miss all the classic work with tracking leatherbacks in the eastern Pacific. See: “Shillinger GL, Palacios DM, Bailey H, Bograd SJ, Swithenbank AM, et al. (2008) Persistent leatherback turtle migrations present opportunities for conservation. PLoS Biol 6(7): e171. doi:10.1371/journal.pbio.0060171”

Likewise you miss all the classic work tracking leatherbacks from the West Pacific. See: “Benson, S. R., T. Eguchi, D. G. Foley, K. A. Forney, H. Bailey, C. Hitipeuw, B. P. Samber, R. F. Tapilatu, V. Rei, P. Ramohia, J. Pita, and P. H. Dutton. 2011. Large-scale movements and high-use areas of western Pacific leatherback turtles, *Dermochelys coriacea*. *Ecosphere* 2(7):art84. doi:10.1890/ES11-00053.1”

Likewise the classic work tracking loggerheads form their largest rookery in the world, see:
“Hawkes et al. Phenotypically Linked Dichotomy in Sea Turtle Foraging Requires Multiple Conservation Approaches. *Current Biology* 16, 990–995. DOI 10.1016/j.cub.2006.03.063”

I appreciate you cannot capture everything, but these are really high-profile studies that have been missed. I could make a list of 50 really important papers that have been missed. I suggest you add some words along the lines of “Our study is a first attempt to compile data from across many different studies and we appreciate that important studies may have been missed through our key word searches. For example, important studies showing pelagic foraging hotspots by leatherback and loggerhead turtles (e.g. Fossette et al, Shillinger et al, Benson et al., Hawkes et al.) were not included”.

Our response: This is a very important comment. We have now carefully reviewed all these papers and managed to extract information on tracks of leatherback sea turtles from Fossette et al 2010, Benson et al 2011, and Shillinger et al 2008.

In the new results, additional foraging areas for leatherbacks are identified in the North Atlantic Ocean. All figures and statistics are now updated to include all identified foraging areas.

We also acknowledge in the discussion section that other important studies, that may identify important foraging areas for sea turtles but not explicitly conform to the methodological criteria of this study, were not included in the analysis (Lines 226-233):

‘The present analysis, by reviewing and mapping available information on tracked sea turtles and by providing a complete and exhaustive mapping of their foraging habitats at the global scale, offers a background for effectively considering critical sea turtles’ hotspots, even in deep sea habitats, in our conservation planning. Our findings could therefore serve as a first step to schedule conservation actions in light of the potential risk of global warming for foraging hotspots. Still, we caution that other significant foraging areas (e.g.,³⁷) exist; nevertheless, the information regarding their distribution does not conform to the methodological criteria applied in this study.’

4th Comments

I suggest you likewise add a couple of lines to say that moving forward it is also important to incorporate foraging ground distribution data from a variety of sources. For example, turtles will not have been tracked from every nesting site. Other sources of important information include surveys on foraging grounds (e.g. aerial) and stranding data and often these can help supplement tracking data to highlight key areas. e.g. see many examples including “A review of the importance of south-east Australian waters as a global hotspot for leatherback turtle foraging and entanglement threat in fisheries. *Marine Biology* 170: 74. <https://doi.org/10.1007/s00227-023-04222-3>”

Our response: Thank you for this valuable suggestion. We clarified that in the revised ms (Lines 233-237):

‘Also acknowledging that sea turtles are not tracked from every nesting site, and that individuals from the same nesting site might visit different foraging areas, it is important to expand our approach to include various sources of information. For example, aerial surveys and stranding data can help supplement tracking data to highlight key areas³⁸.’

5th Comment

Methods. Usually the method to delineate a foraging site is that turtles are tracked to a location where they then remain for many months. I would add this detail to your methods, so it is clear to reader that you did not include incomplete tracks where the transmitter failed before the turtle arrived at its destination.

Our response: Thank you for this comment. We added a sentence to clarify this important aspect. (Line 270-271).

'For our analysis we maintained only complete tracks (n=1035), excluding the cases for which transmitter might have failed before arrival to foraging locations.'

6th Comment

Methods For pelagic foragers how did you assess when the foraging grounds were reached versus animals still migrating ?

Our response: Thank you for the comment. In the revised ms we clearly mention that potential foraging locations of pelagic foragers were extracted following the suggestions of Jonsen et al 2007 (Mar Ecol Prog Ser, 337). The new text reads as (Lines 271-275):

'To identify potential foraging sites for pelagic foragers, we examined shifts in their horizontal behavior. Our focus was on locations where transmitted signals clustered within close proximity and instances where we observed substantial changes in the direction of subsequent transmissions⁴².'

7th Comments

Line 237. "We georeferenced and digitized all areas that were identified as adult foraging sites."

In a line or two I would explain how you did this georeferencing and the likely resolution ... e.g. accurate to a few 10s of km or better ?

Our response: To add clarity we now describe in details how we did the georeferencing in Methods (Lines 275-290):

'The georeferencing process encompassed four main steps. Initially, we gathered the relevant map/s from each article. Subsequently, we pinpointed control points (i.e. distinct and easily recognizable geographic elements) on both the extracted map and a georeferenced map available within the GIS framework. For the third step, these control points served as reference markers to align the extracted map precisely with the georeferenced map in GIS. This alignment process involved meticulous adjustments in position, rotation, and scale to ensure accurate alignment between the two maps. Finally, we attributed coordinates to the extracted map by leveraging the known coordinates from the georeferenced map, utilizing the control points as anchors that connect and synchronize the two maps. Finally, we attributed coordinates to the extracted map by leveraging the known coordinates from the georeferenced map, utilizing the control points as anchors that connect and synchronize the two maps. After digitalizing the tracks, we extracted a minimum of one point to signify a foraging location from each animal track. These points were assigned either at the beginning or end of each trajectory. In instances of complex tracks involving multiple stops or significant directional changes, we selected multiple points. Generally, a foraging point was marked whenever a notable change in trajectory direction was observed in successive transmissions. This method

of selecting multiple points aimed to more accurately delineate the extent of foraging grounds in those tracks.'

And in Discussion (Lines 237-242):

'We further want to emphasize that although our method provides a spatial delineation of key foraging areas, the precision of the derived locations relies heavily on the quality of the gathered data. Specifically, the accuracy of the digitalized foraging points used to identify these areas might fluctuate based on the accuracy of the source maps we employed. Hence, it is imperative for future research to prioritize the use of original tracking data, aiming to mitigate potential spatial biases as much as possible.'

8th Comment

I think you need to discuss that there are many routes to sea turtle conservation. For example, in many countries it will be illegal to hunt, harm or kill sea turtles, regardless of where they are found. So in many countries turtles are protected regardless of whether they are in an MPA. I think you need to make this statement but then explain that while national laws are often in place to protect sea turtles, they may still suffer mortality, e.g. through bycatch or boat strikes. So a holistic view of needed of conservation. It is not just about MPAs. I think this is a key issue that is often ignored.

Our response: Thank you for your suggestion. We have revised the discussion to capture this critical aspect (Lines 211-225):

'Indeed, our analysis showed that the main foraging areas of flatback and hawksbill turtles are located within the neritic zone, possibly justifying why they are relatively better included in existing MPA networks, while the percentages of foraging habitats under protection for the remaining species remain very low or null. However, considering that multiple foraging areas are situated at greater distances from the coast, and tend to expand as we move farther away, we underscore the importance of exploring ways to enhance our conservation efforts in the open ocean. Still, recognizing that many countries have implemented laws to safeguard sea turtles, it's important to acknowledge that populations still face increased mortality risks, such as bycatch or collisions with boats. Despite being protected in these countries, whether within a designated MPA or outside one, sea turtles remain vulnerable. While we appreciate the significance of MPAs in bolstering sea turtle protection, safeguarding this highly mobile species demands a more comprehensive conservation approach. Within this context, delineating key habitats spatially and designing important marine turtle areas becomes imperative.'

9th Comment

In answer to the specific journal questions:

Does the manuscript have technical or conceptual flaws that should prohibit its publication?

No

Are the conclusions original?

yes

Do you feel that the results presented are of immediate relevance for people in your own discipline or for a broader audience?

For a broader audience – see above,

In summary, a nice manuscript that I look forward to seeing published. Graeme Hays

*Thank you very much for your review and opinion on our work!

Reviewer #2 (Remarks to the Author):

Overall, I think this is a nice assessment of the potential change in thermal conditions for sea turtle foraging sites. It's great to have this comprehensive analysis in one study. I think, however, it's a bit of a stretch to refer to this as a 'Climatic vulnerability' when only having discussed thermal conditions. Also need to specify which thermal variables were used in this study. The study is novel in that it encompasses all turtle species and provides a global assessment. It would be more impactful to include additional variables beyond 'thermal conditions' to make this a true study of climate conditions, or include more climate models beyond CM4 to show the variability in the CNo values for each of the foraging sites.

1st Comment

Devising the CNo value is also a nice novel method of making it easier to compare foraging site changes between regions and species. As a result, I think this manuscript will have value to the field.

Our response: We thank you for the positive interpretation of our work. Following your suggestion we now refer to “*thermal*” instead of the more general “*climatic*” term throughout the manuscript and figures. We also added a phrase in the discussion to acknowledge that the inclusion of additional variables and models could enhance the outcomes of this study (Lines 242-245):

‘Finally, the use of other than temperatures, climatic variables, and multiple climatic models could further enhance the outcomes of this study, providing a possible spectrum of climate risk scenarios and enabling further uncertainty analyses.’

2nd Comment

Line 27: Change first sentence to ‘A growing **amount** of evidence...’

Done (Line 28).

3rd Comment

Lines 47 – 49: This sentence is hard to interpret. Please rewrite to clarify.

Our response: We revised the sentence as follows (Lines 48-51):

“While there have been a number of studies where sea turtles have been satellite tracked (e.g., reviewed in¹³), there have been very limited efforts to collate all these accumulated data to examine the overall threats to sea turtles globally.”

4th Comment

Lines 50- 51: This sentence needs a citation.

Added (Miller et al 1997, Arnau Rodriguez, A. 2021) (Line 53).

5th Comment

Line 59 – 61: This sentence needs a citation.

Added (Girard et al 2022, Eckret et al 1999) (Line 62).

6th Comment

Line 63 – 65: This is an oversell of your study. You specifically only worked with temp data (and which temp data needs to be clarified), so to say you examined ‘climate’ stability is inaccurate. You examined temperature stability, and you need to specify which temp (SST, air temp, temp at a depth, ect...).

Our response: Following the reviewer’s suggestion, we now rephrase this sentence as follows (Lines 64-66):

“We also establish an analytical framework to examine the thermal stability of these foraging grounds against sea surface temperature (SST) projections.”

7th Comment

Line 69 – 71: Again, ‘novel climates’ is inaccurate, you mean ‘novel temperatures’. This needs to be changed throughout the manuscript.

Our response: The suggested terms are now used throughout the text. This sentence now reads (Lines 71-72):

“Additionally, our analysis indicates that more than two-thirds of these critical habitats will be exposed to novel temperatures by the end of the century.”

8th Comment

Line 82: It’s unclear what is meant by the ‘surface’ of a foraging hotspot.

Our response: We replaced the term “surface area” by “coverage” (Line 87).

9th Comment

Lines 106 – 108: Again can’t use climatic so loosely, and need to define which temp variables used.

Our response: This term is now changed to thermal and clarified that the temperatures used are SSTs. This sentence now reads (lines 116-117):

“To explore the potential thermal stability of each of the foraging grounds, we used time series of monthly minimum and maximum SSTs”.

10th Comment

Lines 110 – 114: This explanation of the CNo values is repeated from the methods.

Our response: Thank you for pointing this out. In the revised ms, we kept this explanation in the results and retrieved it from methods.

11th Comment

Lines 117 – 138: This is good details, but I think you can also include a little more information on the actual thermal habitats. For each species include the current thermal conditions of their foraging grounds (temp and variability) and then the future conditions to help demonstrate what is meant by high and low novelty of the future conditions. It’s very hard to interpret these CNo results without any actual information about the environments you are referring to.

Our response: To address this and one of the following comments of the reviewer, we conducted a new analysis to estimate the min, max and average SSTs of the foraging areas per species. The detailed information of the methodology and the results per species are now presented in the Supplemental Material. This analysis mainly supports one of our main conclusions that species that forage in tropical areas, thus characterized by higher sea surface temperatures, are generally more likely exposed to novel temperature conditions in the future. However, this analysis is an oversimplification of what is really observed in each one of the species foraging habitats. Indeed, while based on this additional analysis one may draw general conclusions about the species, based on basic descriptive statistics of mean temperatures, the thermal novelty analysis is far more precise and sophisticated, enabling us to identify possible future thermal deviations for each cell inside each foraging area of each species. Moreover, these deviations are not observed based on differences in mean values but rather on differences in the actual temperature

distributions between the baseline and the future time periods, thus providing a far higher statistical power in our results. We now provide these precisions, together with the current and future thermal conditions (average and range) of the foraging areas per species in the Supplemental Material (Table S2). We also refer to this part of the Supplemental Material in the ms (Lines 343-344). Please also see our responses to the reviewer's following comments about the Hellinger Distance and the last comment regarding the information added in the Supplemental Material.

12th Comment

Line 118: Correct to 'olive **ridley** sea turtles'. Use the complete name throughout, not simply 'olive sea turtles'.

Our response: Corrected throughout the text.

13th Comment

Lines 145-147: This sentence needs a citation. Correct spelling of 'Convention'.

Our response: Added (CBD, 2020) (Line 163).

14th Comment

Line 151: You have not actually studied 'suitability', so this sentence is inappropriate based on your results. Either delete or clarify.

Our response: Following the reviewer's suggestion, we rephrased as follows (Lines 166-168):

"We revealed that more than two thirds of the global coverage of the sea turtles' foraging habitats will experience novel sea surface temperatures by the end of century."

15th Comment

Line 151-152: This sentence is worded strangely. Is it meant to say "The tropical zone is the most likely region to have novel climate conditions in the future"? Please clarify.

Our response: We rephrased this sentence following the reviewer's suggestion (lines 168-170):

"The sea turtle species that forage in tropical areas, characterized by higher SSTs, are generally expected of being exposed to novel temperature conditions in the future."

16th Comment

Line 154 – 156: This is assuming northward expansion is the only option for loggerheads. They can also leave the Med. Add a citation to specify your point or clarify this sentence.

Our response: Added (Chatzimentor et al. 2021) (Line 174).

17th Comment

Lines 157 – 158: This sentence needs a citation.

Our response: Added (Patrício et al 2021) (Line 176).

18th Comment

Lines 160– 162: This sentence needs a citation.

Our response: Added (Chatzimentor et al 2021, Patel et al 2021) (Line 180).

19th Comment

Lines 163 – 164: Number this citation.

Our response: Done (Line 182).

20th Comment

Line 188: Sentence ending in this line needs citation.

Our response: In the new version of manuscript, this sentence has been deleted.

21th Comment

Lines 215 – 216: I like this concept of helping to ‘schedule conservation actions’. Perhaps include a figure/table of your recommendations for which foraging site for each species should be protected first.

Our response: Thank you very much for your comment. Our study represents the very first analysis that all foraging grounds of the seven sea turtle species are consistently identified and mapped. It is also the first evaluation of how the sea surface temperatures warming will probably affect these critical for the species areas at the global scale. These elements already provide scientists and managers unprecedented information about the location and possible exposure of sea turtles to changing environments thus providing very valuable elements for setting future conservation actions. However, properly responding to the need of prioritizing actions and measures and identifying which foraging site should be protected first is an analysis apart that goes far beyond the purposes of this study. Acknowledging the importance of such prioritization analyses in the future, we now added a supplementary phrase in the limitation and perspectives paragraph at the end of the discussion section that now reads (Lines 233-237):

“Nevertheless, we emphasize the necessity of climate-smart systematic conservation planning for sea turtles at the global scale. This approach could allow for the incorporation of species complex biology, considering both terrestrial and marine habitats that sea turtles depend on, while accounting for the three-dimensional nature of the ocean environment⁴⁴.”

22th Comment

Lines 236 – 237: You need to provide the citations for all of the publications you used for defining/mapping foraging grounds. Perhaps a supplemental table.

Our response: Thank you for your comment. This is now provided as a Supplementary table.

23th Comment

Lines 284 – 301: Overall the CNo section needs more detail. Why did you choose CM4 vs other climate models? Perhaps worth using a few models with similar resolutions to see if there is variability in calculating the CNo index for each foraging site. You also only incorporate temp, so calling it a ‘Climatic novelty index’ is overselling it. It’s a ‘Thermal novelty index.’ You need to specify which thermal variable was used – SST, air temp, temp at some depth? You cannot simply say ‘thermal conditions’ without specifying which temp variable was used, and then you need to clarify this throughout the manuscript. Need a better definition of the Hellinger Distance. For example, “with CNo \geq 0.5 values indicating some degree of novel thermal conditions in the future” This is far too vague. What is meant by ‘some degree’. It is hard to interpret the change you are trying to demonstrate without some tangible value for this. Is ‘some degree’ equal to a specific temperature range shift? What change in temp is equal to ‘complete novelty’?

Our response: Thank you for indicating these points. In the revised ms, we added more precisions in the methods and throughout the text to provide clarifications about these aspects. We specifically:
1) replaced the term “Climatic Novelty (CNo) index” by “Thermal Novelty (TNo) Index” throughout the ms,
2) specified that thermal conditions refer to sea surface temperatures (SSTs) throughout the ms,
3) to justify the selection of the climatic model we added the following sentence (Lines 338-341):

“In order to couple the identification of foraging areas analysis with the thermal novelty analysis, we specifically required the finest resolution in daily temperature data, and no other climatic model with comparable detail was available.”

Furthermore, we acknowledge this limitation in the Discussion section (Lines 249-252):

“Finally, the use of other than temperatures, climatic variables, and multiple climatic models could further enhance the outcomes of this study, providing a possible spectrum of climate risk scenarios and enabling further uncertainty analyses.”

4) added clarifications about the Hellinger Distance (Lines 353-359):

“The Hellinger Distance is a statistical metric used to quantify the similarity or dissimilarity between two distributions⁵⁵, in this case present and future SST distributions. It is bounded between 0 and 1, with 0 indicating identical distributions, thus no thermal deviation between present and future temperatures and 1 indicating completely disjoint distributions, thus indicating completely novel thermal conditions. A Hellinger Distance value of 0.5 (or higher) generally indicates a moderate dissimilarity between the two distributions.”

We also replaced the term “some degree” by “moderate degree” throughout the text.

24th Comment

Fig 1, Fig 3, Fig S1, and Fig S2: For all maps, perhaps crop out Antarctica and portions of the Arctic and zoom in a bit.

Our response: We appreciate this suggestion, however, the resolution and appearance of the foraging grounds is not significantly improved, so we preferred to present the distribution of foraging grounds in the broader worldview. Nevertheless, we are willing to make any graphical modifications that the editorial board would judge most suitable.

25th Comment

I think an interesting additional set of maps would be to show the CNo values for each foraging ground by species. This can be supplemental info, but I think it would be helpful for researchers who are more focused on certain species over other. Having these maps would be very helpful in providing more in depth information for Fig3a and b.

Our response: Thank you for this suggestion. We now added the thermal novelty (TNo) values per foraging area for each of the seven species in separate maps in the Appendix (see Figure S5). We refer to this figure in the ms (Lines 153-154).

26th Comment

Table S1: Perhaps include some summary of the ‘thermal conditions’ (again is this SST, air temp, temp at some depth) as separate columns – min, max, average, time of year.

Our response: Following the reviewer’s suggestion, we included an additional table in the Supplemental Material where we note the minimum, maximum and average temperatures (SSTs) per species for both the baseline (2000-2014) and the future period (2085-2100) (See Table S2).

*Thank you very much for your review and opinion on our work!

REVIEWERS' COMMENTS:

Reviewer #1 (Remarks to the Author):

The authors have made a good effort to revise the manuscript in line with my previous comments. I have a few remaining minor points.

1. Figure 2. What are all the different colours ? I could not understand the figure. The legend is poor. Give more details in the legend.
2. Lines 17. I would say "n=1035 INDIVIDUALS".
3. References 16 and 38 are the same.

I look forward to seeing the work published.

Reviewer #2 (Remarks to the Author):

Great job updating the manuscript. I think it is ready for publication. I have one small suggestion, please add a legend to the supplemental maps of the separate species for TNo colors.

Response to Reviewer

Reviewer #1

1st Comment

Figure 2. What are all the different colours ? I could not understand the figure. The legend is poor. Give more details in the legend.

Our response: Thank you for pointing this out. We modified the caption of Figure 2 and add the title above pie chart. The revised caption now reads:

“Fig. 2. The percentage coverage of sea turtles’ foraging grounds by the current network of marine protected areas (MPAs) per species (bars) and for all turtle species combined (pie chart). Less than 2% of global foraging grounds are covered by MPAs.”

2nd comment

Lines 17. I would say “n=1035 INDIVIDUALS”.

The word ‘individuals’ is added (Line 17).

3rd comment

References 16 and 38 are the same.

Thank you for pointing this out. We merged these same citations and cited them as one.

I look forward to seeing the work published.

*Thank you very much for your review and opinion on our work!

Reviewer #2

Great job updating the manuscript. I think it is ready for publication. I have one small suggestion, please add a legend to the supplemental maps of the separate species for TNo colors.

Thank you for your suggestion. We added the legends to supplemental Figure 2.

*Thank you very much for your review and opinion on our work!